# DOSA: One-Loop DSE for DNN Accelerators Using Differentiable Models

Charles Hong*, Qijing Huang†, Grace Dinh*, Yakun Sophia Shao*,

*University of California, Berkeley. *charleshong@berkeley.edu,dinh@berkeley.edu,ysshao@berkeley.edu*
†NVIDIA. *jennyhuang@nvidia.com*

*Abstract*—The process of hardware design space exploration requires both hardware parameters and mappings from the algorithm onto the target hardware to be discovered and optimized. Previous work has largely approached this simultaneous optimization problem by separately exploring the hardware design space and the mapspace—both individually large and highly nonconvex spaces—independently. The resulting combinatorial explosion has created significant difficulties for optimizers. We introduce DOSA, which consists of differentiable latency and energy models, as well as a gradient descent-based optimization technique to simultaneously explore both spaces and identify high-performing design points. Experimental results demonstrate that DOSA outperforms random search and Bayesian optimization by 2.80× and 12.59×, respectively, in improving DNN model energy-delay product, given a similar number of samples. In particular, we demonstrate DOSA's modularity and flexibility via transfer to a real DNN accelerator setting, where we achieve a 2.07× improvement in energy-delay product by augmenting our analytical model with a learned model.

## I. INTRODUCTION

To develop efficient and high-performance DNN accelerators in a fast and cost-effective manner, automated design space exploration (DSE) has emerged as a powerful technique. The hardware DSE flow [15], [20], [26] involves optimizing over two search spaces: the *hardware design space*, which describes hardware design parameters such as interconnect topology and buffer and systolic array sizes, and the *mapspace*, which describes how applications are executed on the target hardware and encompasses decisions such as loop tile dimensions, dataflow, and spatio-temporal mapping.

Both the hardware design and mapping spaces are vast, high-dimensional, and comprised of both categorical and discrete variables. Furthermore, evaluating the performance of a hardware configuration and a mapping can be computationally expensive. The size of the combined optimization space and the cost of evaluating points in it pose formidable challenges to DSE algorithms.

Much prior work [6], [17], [23], [26], [30], [33] has approached this problem using *hardware-first search*. These methods directly search over the space of possible hardware configurations. The performance of each hardware configuration is calculated by first constraining the mapspace to mappings that are compatible with the hardware configuration, then optimizing over the constrained (highly discontinuous) mapspace. In most cases, the mapspace optimization is done iteratively, rendering this process a *two-loop approach* iterating

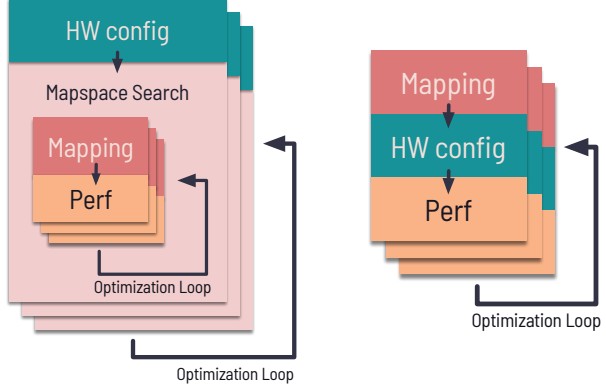

Fig. 1. Hardware-first, two-loop (left) and mapping-first, one-loop (right) DSE approaches.

over both the hardware space and mapspace. As a result, these approaches must contend with a combinatorial explosion of possible configurations.

Alternatively, *mapping-first* approaches, as proposed in [10], [31], optimize primarily over the mapspace. For each mapping, optimizing over the hardware design space is a straightforward process consisting of finding the minimal hardware configuration capable of supporting the mapping. As a result, the loop for hardware search is eliminated, allowing the entire DSE process to be encapsulated in a single loop. Furthermore, the lack of hardware resource constraints also significantly simplifies the mapspace search problem.

Despite these advantages, mapping-first approaches must still contend with the size of the mapspace and the nonconvexity of the performance over this space. Prior works have either directly applied black-box optimization methods [10], [25], which rely on a large number of samples, or pruned the search space using architecture-specific heuristics [31], leaving a large set of candidate design points for the architect to select from.

Rather than optimize a black box, we can leverage the domain knowledge provided by white-box performance models like Timeloop [20], [28], which can provide precise feedback quickly enough to fully automate the design space exploration process. This paper follows this approach, using performance models as an optimization target for mapping-first search. Specifically:

- We build closed-form *differentiable* and interpretable

| | Name | Mapspace Search | Hardware Search |
|---|---|---|---|
| **Two-loop Searchers** | Spotlight [23] | BB-BO | BB-BO |
| | VAESA [6] | ILP [7] | VAE+BB-BO/GD |
| | FAST [33] | BB-LCS [12] + ILP | BB-LCS |
| | HASCO [30] | BB-RL | BB-BO |
| | NAAS [17] | BB-ES | BB-ES |
| | MAGNet [26] | Heuristics | BB-BO |
| **One-loop Searchers** | DiGamma [10] | BB-GA | |
| | Interstellar [31] | Heuristics | |
| | **Our work: DOSA** | **GD** | |

performance models for latency and energy on DNN accelerators. Our models are as precise as the program-based analytical models, while also being amenable to white-box optimization using gradient descent.

- We introduce a DNN model to predict the variation between analytical model and real hardware accelerator performance, augmenting our interpretable analytical models for real hardware DSE.
- We then introduce DOSA, a mapping-first one-loop DSE flow that uses gradient descent to find the most efficient hardware parameters and mappings for target multi-layer DNNs; to the best of our knowledge, this is the first work to perform mapping-first DSE to multi-layer neural nets. DOSA converges at least 40% faster than state-of-the-art DSE approaches.
- We benchmark our results on the Gemmini accelerator, showing a 2.07× EDP improvement over hand-designed configurations.

## II. BACKGROUND

Hardware DSE typically includes two key optimizations: the mapping search and the hardware search. To address the mapping complexity for DNNs, many DNN compilers [1]–[3], [13], [19], [21], [22] and accelerator-aware mapping techniques [5], [7], [8], [16], [20], [31] have been developed. In addition, there has been extensive research in the area of hardware parameter search [14], [32].

### A. Optimization Techniques

In recent years, there has been a growing body of research focused on tackling the compounding search space of mapping and hardware designs with the goal of achieving higher hardware efficiency and lower development costs.

Optimization techniques used in this search process can be broadly categorized into three types: heuristics, black-box optimization (BB), and white-box optimization. Heuristics involve using domain-specific knowledge and experience to guide the search process and reduce the size of design space. In contrast, BB relies on sampling and machine learning techniques to leverage the characteristics of the problem derived from sampled data in order to find the optimal solution. In white-box optimization, the relationship between the optimization variables and the objectives is known and captured in mathematical models. Numerical optimization techniques

like linear programming (LP) and mixed-integer programming (MIP) can be used if the relationship can be expressed in specific frameworks. Gradient descent (GD) techniques can be applied if the relationship can be expressed in a differentiable expression. Compared to black-box optimization, white-box (WB) optimization is generally more efficient as it can exploit the known objective model to guide the optimization process, resulting in faster convergence. However, it requires the objective model to be known and accurately specified.

### B. Co-exploration Frameworks

As shown in Table I, most prior work [6], [17], [23], [26], [30], [31], [33] treats the mapping and hardware co-search as a two-loop process and applied a combination of various optimization techniques to address each search space independently. While independently applying optimization techniques to the mapspace and hardware space can be effective, the two-loop searchers can be susceptible to combinatorial explosion, as the vast search space multiplies the number of potential options for mappings and hardware parameters together.

To reduce the size of the compounding search space, one-loop searchers, such as DiGamma [10] and Interstellar [31] have been proposed. Single-loop search tackles the co-search problem from a mapping-first approach that infers the minimal hardware requirement from hardware-agnostic high-performance mappings found in single-loop mapping search. In such approaches, the hardware DSE space is similar in size to the mapping space. However, DiGamma employs BB-GA which treats the mapping performance as a black-box and needs to evaluate many unique hardware and mapping configurations iteratively to achieve a good mapping and hardware design. Interstellar, on the other hand, only explores a limited space of pre-selected mappings and as a result only a limited space of hardware design is evaluated.

Unlike previous one-loop approaches that rely on black-box optimizations or heuristics, DOSA takes a novel approach by formulating the analytical performance and energy model in [20] as a differentiable white-box model. DOSA uses gradient descent to optimize the mapping variables in the direction of steepest descent of the EDP objective function on the mathematical model. This allows DOSA to explore a comprehensive set of mappings and efficiently generate high-quality hardware and mapping configurations without the need for sampling from simulators extensively.

### III. DOSA OVERVIEW

This paper presents DOSA, a one-loop differentiable-model-based DSE framework to optimize the mappings and hardware simultaneously for target DNN models. DOSA captures key relations between DNN mapping factors and performance objectives in a differentiable analytical model. In addition, DOSA introduces a data-driven DNN model to capture the performance variations between analytical model and real hardware. By applying white-box optimization to the model and calculating the hardware parameters using minimal parameterization, DOSA achieves high-performance accelerator

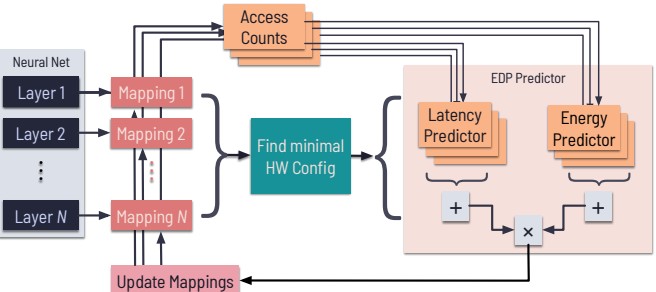

Fig. 2. An architecture diagram of DOSA.

```
// Layer:
// N=1, R=1, S=1, P=56, Q=56, C=64, K=64

// DRAM (Weights: 4096 Inputs: 200704 Outputs:
200704)
for p3 in [0:56]:
  for q3 in [0:4]:
    // Scratchpad (Weights:4096, Inputs: 896)
    spatial_for k2 in [0:64]:
      // Accumulator (Outputs:896)
      spatial_for c1 in [0:64]:
        // Registers (Weights: 4096)
        for q0 in [0:14]:
```

Per-Layer Min-HW

Scratchpad: (4096 + 896) words x 1 B/word ≈ 5 KB

PEs: 64x64

Accumulator: 896 words x 4 B/word ≈ 4 KB

Gemmini

$max(\cdot)$

Fig. 3. Mapping to hardware parameters conversion in DOSA.

design and mapping while significantly reducing the time and costs associated with DNN accelerator DSE.

**Toolflow.** Figure 2 shows how DOSA simultaneously optimizes mappings and hardware for a given workload consisting of a set of layers. DOSA iteratively updates mappings using gradient descent, finding minimal hardware requirements at each step. Specifically:

1) *Initialize the search.* We first select a random valid hardware parameters and use CoSA [7], a one-shot optimization-based mapper, to map our set of target DNN layers onto it.

2) *Find minimal hardware parameters.* We compute the hardware resource requirements of each layer-wise mappings and set hardware parameters to support all mappings.

3) *Compute EDP cost from current mapping and hardware parameters.* We first compute the number of accesses made by each mapping to each memory level in the accelerator using the differentiable model in DOSA. These access counts are combined with our current set of hardware parameters to generate roofline-based latency predictions and event-based energy predictions for each layer's mapping, from which we derive a single EDP prediction.

4) *Update all mappings* in parallel using gradient descent.

5) *Repeat from Step 2.*

The components that make up this flow are detailed in the following sections.

## IV. DOSA DIFFERENTIABLE MODEL

Given the absence of a differentiable, analytical model for DNN accelerators in current literature, we present our approach for constructing such a model that achieves accuracy on par with Timeloop in our problem space. To account for performance variations in real hardware that are difficult to capture and express in analytical models, we additionally train a differentiable DNN model to futher improve the accuracy of the performance model.

We target the open-source DNN accelerator Gemmini [4], whose most notable architectural components are 1) a systolic array of processing elements (PEs), 2) accumulator SRAM, 3) scratchpad SRAM, and 4) DRAM. Specifically, we target the weight-stationary (WS) configuration of Gemmini.

We evaluate Gemmini at two levels of fidelity: Timeloop and RTL. We use Gemmini-TL to refer to the Timeloop definition of an accelerator analogous to Gemmini, evaluated with an architectural model, and Gemmini-RTL to refer to the RTL implementation of Gemmini available at https://github.com/ucb-bar/gemmini, evaluated using cycle-accurate RTL simulation.

### A. Computing Hardware Resource Requirements

As depicted in Figure 3, the capacity requirements at each level are first computed. Then, we take a parameter-wise max to generate a design that will support all current mappings. The exact formulas used are not enumerated here, but its accuracy relative to Timeloop is demonstrated in Section IV-E.

*1) PE Capacity Requirements:* Gemmini supports only square arrays of processing elements. In its WS (weight stationary) configuration, it can parallelize the input channel and output channel dimensions, each along one side of the array. Hence, we need to configure a square PE array that is large enough to accommodate the square of the larger of these two spatial factors.

*2) Buffer Capacity Requirements:* Buffer capacities required at a given level for the weight ($W$), input ($I$), and output ($O$) tensors are computed from the spatial and temporal tiling factors at and inner to that level. The total buffer capacity requirement at a given memory level is the sum of sizes of each tensor stored at that level.

### B. Traffic Estimation

To capture the performance of the accelerator, we utilize differentiable, but non-convex functions to model the writes, updates, and reads to each buffer level. These values are computed per mapping.

### C. Latency Modeling

We calculate the latency cycles required for compute by dividing the total number of multiply-accumulate (MAC) operations in a layer by the product of all spatial factors in

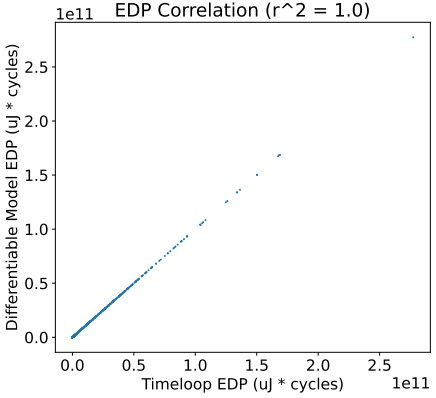

Fig. 4. Correlation of DOSA differentiable model with Gemmini-TL.

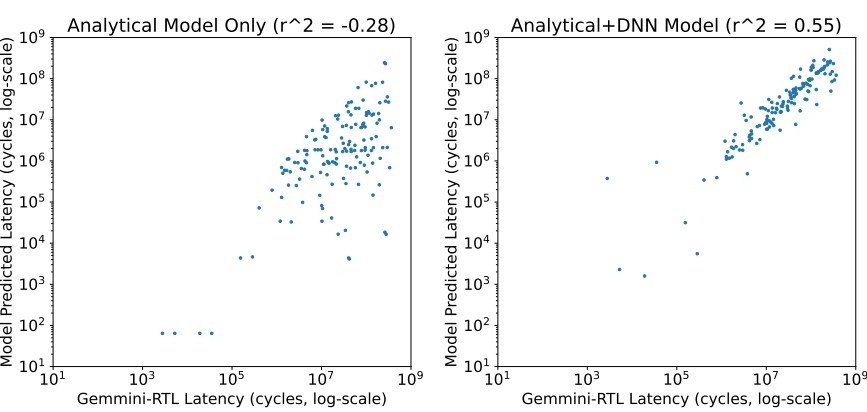

Fig. 5. Accuracy of Gemmini-RTL latency modeling with and without DNN augmentation.

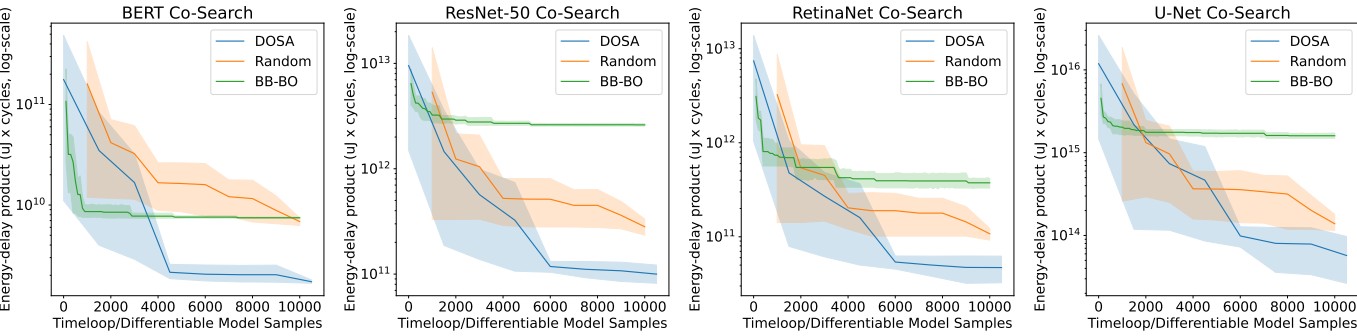

Fig. 6. DOSA EDP optimization of Gemmini-TL on 4 distinct workloads, versus baselines. Each line represents the mean (across 5 runs) best point found after $x$ model evaluations. The shaded regions represent a 95% confidence interval across the 5 runs.

a mapping (i.e., the number of parallel processing elements utilized). To compute memory access latency, we divide the total number of memory accesses by the memory bandwidth. We calculate the memory latency for each memory level $i$ utilized in Gemmini, including accumulator SRAM, scratchpad SRAM, and DRAM. We consider the maximum latency among all memory levels and the compute as the final latency since performance is limited either by memory or compute. The latency formulations are provided below:

$$
\begin{aligned}
\text{Compute\_Latency} &= \frac{\text{\# of MACs in Layer}}{\prod(\text{Spatial Factors})} \\
\text{Accesses}(i) &= \text{Reads}(i) + \text{Updates}(i) + \text{Writes}(i) \\
\text{Mem\_Latency}(i) &= \frac{\text{Accesses}(i)}{\text{Bandwidth}(i)} \\
\text{Mem\_Latency} &= \max_i(\text{Mem\_Latency}(i)) \\
\text{Latency} &= \max(\text{Compute\_Latency} \\
&\quad, \text{Mem\_Latency})
\end{aligned}
$$

### D. Energy Modeling

Energy is modeled via data collected for a 40nm process using Accelergy [28] and CACTI [18]. In our model, compute, register access, and DRAM access energy are constant per word, whereas SRAM access energy per word scales with the number of SRAM rows and columns.

$$
\begin{aligned}
\text{Energy}(i) &= \text{Accesses}(i) \times \text{EPA}(i) \\
\text{Energy} &= \text{MACs} \times \text{EPA}_{PE} + \sum_{i \in M} \text{Energy}(i)
\end{aligned}
$$

### E. Modeling Gemmini-TL

To demonstrate that our differentiable performance model does not compromise accuracy relative to other analytical models, we conducted experiments to show that DOSA has equivalent accuracy to Timeloop [20] for an accelerator analogous to Gemmini. We compare 100 random Gemmini configurations, 73 matrix multiplication and convolutional layers, and 10 random mappings per layer. Figure 4 demonstrates that the latency, energy, and EDP results from our differentiable model correlate almost perfectly with the results from Timeloop.

### F. Modeling Gemmini-RTL

In general, analytical models do not completely capture hardware performance [27], [29]. Variations caused by specific implementation details and complex hardware-software interactions may be unknown to the designer or difficult to capture mathematically. One potential solution is to augment analytical models with learned surrogate models. Since many learned models, such as deep neural networks or polynomial regression models, are differentiable, DOSA is particularly well-suited to work with such models.

In this case, we train a deep neural network to predict the difference between our analytical model's latency predictions for a layer and the real latency of Gemmini-RTL, evaluated using FireSim [11]. The model's inputs include the layer's dimensions, a mapping, and a hardware configuration. The inputs are also augmented with the roofline latencies computed in the analytical model. The model's architecture is similar to the that of the model used in Mind Mappings [5]. It contains 7 hidden fully-connected layers and a total of 5737 parameters.

## V. DOSA OPTIMIZATION

Constructing a differentiable performance model allows DOSA to optimize hundreds of parameters (tens per layer, times tens of layers) at once using gradient descent (GD). Differentiability is implemented using PyTorch automatic differentiation. GD start points are generated via random hardware configuration, plus CoSA [7] mappings. The GD loss term is the total performance metric, for example energy-delay product (EDP). We compute the EDP of a full model by summing each mapping's latencies and energies, then multiplying the resulting sums.

**Loop Ordering.** Notably, the loop ordering term of the mapping is not differentiated in our formulation. The loop ordering of each layer is shuffled every time mappings are rounded to the nearest valid mapping, and the best differentiable model-predicted loop ordering is selected. We select between 3 loop orderings per layer, per level: weight-stationary, output-stationary, and input-stationary. As noted by works such as AIRCHITECT [24], it is possible to accurately analyze problem dimensions and buffer sizing to select the optimal dataflow. Others have noted that dataflow and loop ordering are much less impactful than tiling parameters in the mapping problem [9], [31]. This makes the hardware-mapping co-design problem more amenable to gradient descent. Another way to optimize non-differentiable terms is to add a neural network-based surrogate model that can propagate a gradient to these terms, as shown later in this work.

**Start Point Rejection.** In subsequent iterations of start point generation, a start points is rejected, and a new hardware configuration is selected, if its differentiable model-predicted performance is more than $10\times$ that of the best start point seen thus far.

**Rounding.** Since gradient descent may result in non-integer tiling factors, before any mapping is evaluated, it is rounded to the nearest valid mapping by rounding each tiling factor to the nearest divisor of its corresponding problem dimension, subject to the constraint that the rounding process does not cause the product of tiling factors for that dimension to exceed the total problem size. This process iterates from the innermost to outermost memory level.

**Preventing Exploration of Invalid Mappings.** We do not include tiling factors at the outermost (DRAM) level as optimization targets, and instead infer them by dividing the total problem size at each dimension by the product of the rest of the tiling factors for that dimension. In order to prevent

exploration outside the space of valid mappings, we add a loss term for DRAM tiling factors less than 1.

## VI. EVALUATION

### A. Experimental Setup

We compare the performance of DSE algorithms on a variety of target DNN models that can handle a diverse set of tasks, such as natural language processing, image classification, object detection, and image segmentation. The hardware parameters we select using the capacity requirement calculations in Section IV are PE dimensions, accumulator SRAM sizing, and scratchpad SRAM sizing. PE dimensions come from spatial tiling factors, which can be directly used as they are always positive integers. PE array size is capped at 128x128. SRAM sizes are rounded up to increments of 1 KB. For these experiments, the specific descent algorithm DOSA uses is Adam, an optimizer similar to gradient descent with momentum. Rounding happens every 500 steps and GD is run for 1490 steps on each start point, unless otherwise noted.

We set up CoSA with equally partitioned scratchpad for inputs and weights. Our Bayesian optimization-based hardware-mapping optimizer baseline is similar to Spotlight [23]. This is a two-loop method which trains a Gaussian process model with 100 hardware designs and 100 mappings per layer per hardware design, and uses this model to selected the hardware design and mappings with the best predicted performance from 10000 candidates per problem.

### B. Hardware-Mapping Co-Search Performance

Our evaluation finds that DOSA is able to identify significantly more performant co-design points than either random search or Bayesian optimization with a similar number of samples. BB-BO uses Timeloop simulation as a black-box optimization metric for Gemmini-TL. The random search- and DOSA-generated co-design points are also evaluated under this setup. After around 10,000 model evaluations, the geometric mean of EDP improvements for DOSA versus random search is $2.80\times$, and $12.59\times$ versus BO. Evaluations done using Timeloop are considered equivalent to evaluations done using DOSA's differentiable model.

### C. Gemmini-RTL Optimization with DOSA

In this section, we assess the efficacy of our one-loop differentiable-model-based gradient descent approach for real hardware design. We also enhance the analytical model with a DNN model to narrow the performance gap between analytical model predictions and actual hardware performance.

After modifying DOSA latency predictors to include the learned latency model, we run gradient descent and generate a predicted optimal set of mappings and buffer sizes for 16x16 PE Gemmini-RTL, fixing PE dimensions and adjust only buffer sizing and mappings. We compare the performance of DOSA-generated mappings to the mappings generated by the Gemmini-RTL default heuristic-based mapper, and the default scratchpad and accumulator sizings of 128 KB and 32 KB respectively. We run DOSA in two settings. First, we utilize the

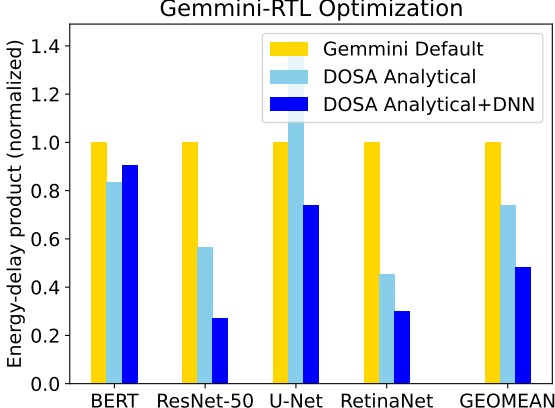

Fig. 7. DOSA augmented with a learned Gemmini-RTL performance model ("DOSA Analytical+DNN") finds more performant hardware-mapping co-design points than Gemmini's default hardware and mapper or DOSA with analytical model only ("DOSA Analytical").

original formulation from Section IV to generate scratchpad and accumulator sizings, along with corresponding mappings, for each workload. Second, we replace the original analytical model-based latency predictor with the DNN-augmented version. We initialize gradient descent with Gemmini-RTL default buffer sizings, plus CoSA mappings. Over the four test workloads, which are not included in the training data for the latency predictor, the analytical-only version of DOSA achieves a $1.35\times$ EDP improvement, and the Gemmini-RTL trained version of DOSA yields a $2.07\times$ improvement, as shown in Figure 7.

## VII. Conclusion

In this paper, we present DOSA, a model-based approach mapping-first DSE. By constructing a differentiable analytical performance model for a DNN accelerator, we can use gradient descent to perform an efficient one-loop co-search of both the hardware and mapping spaces. This enables us to to perform DSE targeting *multi-layer* neural net workloads, attaining an EDP $2.80\times$ better than random search and $12.59\times$ better than Bayesian optimization, while using a similar number of samples.

DOSA demonstrates that interpretable, designer-trusted architectural modeling and ML-based optimization methods are not necessarily mutually exclusive, and in fact can be combined to improve the accuracy of performance models and the convergence of DSE. The modular construction of our performance model enables DOSA to be more easily extended to different performance objectives than existing performance modeling and optimization methodologies. We demonstrate this principle by pairing our analytical latency model with one experimentally trained on a RTL simulation and event-based energy analysis of Gemmini, improving EDP by $2.07\times$ over the default Gemmini configuration.

This *modular* approach to building and combining performance models and using different sources of performance data for DSE suggests an avenue of attack for optimizing

objectives that are expensive to compute (e.g. fine-grained performance simulations) where prior black-box approaches may face challenges. With this work, we move one step closer to bridging the gap between architectural models and real silicon.

### Acknowledgements

This work was supported in part by the UC Berkeley SLICE Lab industrial sponsors. We thank Hasan Genc and Divija Hasteer for their help with Gemmini, and Xiangyu Xu and Jonathan Wang for their help generating baseline data.

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
