# OpenReview forum: "DOSA: One-Loop DSE for DNN Accelerators Using Differentiable Models"
_iscaconf.org/ISCA/2023/Workshop/ASSYST — ASSYST Oral_

### Official Review · Reviewer_jHiW · 2023-05-05
**This paper proposes an interesting idea about how to formulate the  co-optimization of DNN accelerator hardware and mapspace with a differentiable performance model. With such a differentiable model, the authors can use gradient descent to efficiently and effectively to simultaneously explore both spaces and identify high-performing design point.**

**Rating:** 5
**Confidence:** 4

**Review:**

The research targets an interesting problem. I am sold on the desire for a differentiable performance model to enable efficient design search. However, there are also several issues that could be addressed.

Firstly, the limitations and general feasibility of creating a differentiable performance model could be discussed more thoroughly. It is also not clear to me what does the differentiable performance model enables exactly, more efficient search or better search. Evaluations with a large number of iterations for the baselines may give us better answers here.

Additionally, the target set for the research is quite fixed and it is not clear whether the differential condition still holds in more complicated hardware design space.  The deal with loop ordering where the differential condition does not hold seems to be ad-hoc to me. The evaluation of the research could be improved significantly.



**Review (Strengths/Weaknesses):**

Strength:
+ An interesting research problem
+ A differentiable performance model is desirable for efficient design search.

Weakness:
- The limitations or general feasibility of making the performance model differentiable can be better discussed.
- The target setting is a quite fixed hardware dataflow setting, which somehow put heavy constraints on what they proposed in the first place.
- The evaluation can be significantly improved.


**Reviewer Expertise:**

Knowledgeable: I used to work in this area and/or I try to keep up with the literature but might not know the latest developments.

---

### Official Review · Reviewer_UMJo · 2023-05-05
**A good paper on hardware-mapping co-exploration.**

**Rating:** 7
**Confidence:** 4

**Review:**

The paper proposes a differentiable analytical model-based DSE framework for DNN accelerator hardware and mapping co-optimization. The proposed framework shows significant improvement in accelerator performance as compared to prior arts.

Pros: 1. The method is fast and does not require time-consuming training procedures;
2. The method shows great improvement in accelerator performance as compared to prior methods, such as BO.
3. The method also targets co-optimization while prior methods mainly focus on single optimization, e.g., mapping or hardware.
4. Writing is also good.

Cons: 1. Minor typos, such as "which defines" (repeating two times) in Section III. B.
2. Lack of some technical details, such as the searching grid of each parameter.

**Review (Strengths/Weaknesses):**

See my comments above.

**Reviewer Expertise:**

Knowledgeable: I used to work in this area and/or I try to keep up with the literature but might not know the latest developments.

---

### Official Review · Reviewer_o3yB · 2023-05-07
**OPT: Differentiable Model-Based DNN Accelerator Hardware and Mapspace Co-Optimization**

**Rating:** 5
**Confidence:** 3

**Review:**

This paper presents an analytical approach to select hardware design space and DNN execution mapping. The approach defines the problem as a set of differentiable equations that can be optimized with gradient descent. The authors show that this approach is significantly better than random search and Bayesian optimization.

**Review (Strengths/Weaknesses):**

Strengths:
* The authors formulate the optimization in a well constructed set of equations that accounts for memory, compute and power.
* The paper does a great job in summarizing and comparing related works.
* Though I am unsure whether the approach is novel, it is refreshing to see a differentiable methodology applied to hardware/software codesign.

Weaknesses:
* The approach used in the paper focuses solely on optimizing convolutions and matrix multiplications on a given template architecture (Gemmini). Vector operations, data manipulation, on-chip / off-chip interconnects are also important components of any DNN accelerator. It is unclear if the differentiable approach presented in the paper can be applied more broadly to cover the design of those units, or to a more significantly different architecture (e.g. GPU-like, dataflow, CGRA).
* The evaluation section of the paper is thin. For example, some detailed comparisons and analysis of the best hardware/mapping configuration chosen by GD/Random/BO approaches would be insightful, particularly since the author argues that the OPT approach is “interpretable”.
* It is surprising that the BO curve in Figure 2 is almost flat across all models. Insights on why this is the case would make the paper stronger.


**Reviewer Expertise:**

Little or no familiarity.

---

### Official Review · Reviewer_xa9J · 2023-05-07
**Good paper with a lot of potential**

**Rating:** 6
**Confidence:** 4

**Review:**

The paper suggests using differentiable methods, called OPT, built on top of analytical models. With this approach, this work co-optimizes hardware and mapspace in tandem. The results show that OPT outperforms conventional methods (random search, bayesian optimization) across a range of machine learning models, including recent language models such as BERT.

**Review (Strengths/Weaknesses):**

**Strengths**

(1) Proposing a holistic analytical solutions combined with gradient-based search method is very promising and beneficial in rapid evaluation of design points.

(2) Evaluating across a range of models from ResNet to recent BERT-based models.

**Weaknesses**

(1) The paper argues that the technique only works for square array, but it was not clear from the texts why this is the case. Whether it is a limitation of the analytical model or an inherent limitation of the approach (I assume the former). Additional clarification would help.

(2) There were multiple ad-hoc decisions that was not clear to me. Only selecting the points that are 10$\times$ better that the best design point seen so far. Rationalizing about these decision or ablation studies would help to better understand the point.

(3) While intuitively the proposed solution seems to be more sample-efficient, but running meticulous hyperparameter search for other techniques are critical. I can see that BO seems to saturates very quickly, something that may be possible to escape with better hyperparameters (or maybe I am missing something that is fundamentally limiting with BO).

** Minor **
- Page 3, repeated words `which defines which defines which`.

**Reviewer Expertise:**

Expert: I have written one or more papers on this topic and/or I currently work in this area.